# Analysis of the Influence of the Motion State of Ultra-Thin Sapphire Based on Layer-Stacked Clamping (LSC)

**DOI:** 10.3390/mi14061124

**Published:** 2023-05-26

**Authors:** Zhixiang Chen, Shunkai Han, Ming Feng, Xianglei Zhang

**Affiliations:** 1College of Mechanical and Electrical Engineering, Wenzhou University, Wenzhou 325035, China; hanshunkai@stu.wzu.edu.cn (S.H.); fming@wzu.edu.cn (M.F.); zhxile512@wzu.edu.cn (X.Z.); 2Xinchang Zhejiang University of Technology Science and Technology Research Institute, Shaoxing 312500, China

**Keywords:** motion state, layer-stacked clamping, sapphire wafer, double-sided polishing

## Abstract

Ultra-thin sapphire wafer processing is receiving increasing attention in the LED substrate industry. In the cascade clamping method, the motion state of the wafer determines the uniformity of material removal, while the motion state of the wafer is related to its friction coefficient in the biplane processing system, but there is little relevant literature discussing the relationship between the motion state of wafer and friction coefficient. In this study, an analytical model of the motion state of sapphire wafers in the layer-stacked clamping process based on the frictional moment is established, the effect of each friction coefficient on its motion is discussed, the base plate of different materials and different roughness are experimentally studied, the layer-stacked clamping fixture is prepared in this way, and finally the failure form of the limiting tab is analyzed experimentally. The theoretical analysis shows that the sapphire wafer is mainly driven by the polishing plate, while the base plate is mainly driven by the holder, and the rotation speed of the two is not the same; the material of the base plate of the layer-stacked clamping fixture is stainless steel, the material of the limiter is glass fiber plate, and the main form of failure of the limiter is to be cut by the edge of the sapphire wafer and damage the material structure.

## 1. Introduction

Sapphire wafers have excellent material properties, stable chemical characteristics, and a high dielectric constant, making them an excellent choice for hybrid substrates [1,2,3]. However, the high hardness and significant chemical inertness of sapphire single crystals make them a typical difficult-to-process material [3,4,5].

Double-sided polishing technology is the preferred method for processing sapphire wafers [6]. Many scholars have conducted research on the double-sided CMP processing of sapphire wafers. In the study of double-sided trajectories, Hu et al. established a double-sided planetary grinding mathematical model [7] and evaluated the nonuniformity of the lapping wheel [8,9]. Zhang proposed the specific detailed distribution ranges of process parameters by the trajectory uniformity of the double-sided mechanical polishing [10]. Yang proposed a method to measure the workpiece rotational speed and established the mathematical model considering the workpiece rotational speed through the track point distribution (NUTPD) to evaluate the trajectory uniformity [11]. Wang demonstrated a method of trajectory in double-side autonomous grinding considering the dynamic friction coefficient [12]. Deng et al. revealed the influence of polishing process parameters on the surface quality of sapphire [13] and then obtained and experimented on the sapphire DS-CMP processing plan based on the Box–Behnken design [6], given the optimal process parameter combinations, and obtained the maximum MRR of 387.59 nm min^−1^. Satake, based on the pressure distribution control, proposed an analytical model of the removal distribution [14]. Hashimoto developed an identification method of the friction coefficient to accurately estimate the wafer behavior [15] and proposed a calculation method for workpiece profile variation during double-sided lapping, concluding that uneven workpiece thickness can be modified by the process [16]. Hirose and Satake developed a method for optimizing the operating conditions of a double-sided lapping machine [17,18]. The process was also theoretically investigated to elucidate and improve the complex process. Some scholars have studied laser-assisted grinding and self-rotating grinding of GaN crystals, and the research results show that laser-assisted grinding with an appropriate laser power density reduced the subsurface damage depth and wear damage of the abrasive [19], obtaining an ultra-smooth surface of 1 nm [20], which provides a solution for the efficient processing of sapphire wafers.

Due to the poor thermal conductivity of sapphire materials [21], the temperature rise of the LED active area will cause the sapphire substrate to be heated, which will have a fatal impact on the light output characteristics and service life [22]. Therefore, it is usually necessary to thin the sapphire wafer to 100 μm when preparing the LED substrate [23,24]. However, due to the difficulty of achieving good strength and rigidity with an overly thin planetary wheel holder, it is impossible to achieve low-cost ultra-thin flat part processing. To this end, the authors’ research group proposed a layer-stacked clamping method (LSC) [25] and conducted experimental research on the adsorption mechanism of the layer-stacked clamping method in previous studies.

Based on the LSC method, this article analyzes the force state of the layer-stacked fixture and sapphire wafer, and studies the motion state of the sapphire wafer and the base plate and its influencing factors through experimental research on the friction coefficient between sapphire wafers and base plates of different materials and roughness, combined with theoretical analysis to determine the materials of the base plate and limiter, and through experimental analysis of the failure form of the limiter under clamping conditions.

## 2. Motion Model of Workpiece

Under the LSC double-sided processing method, the workpiece only generates self-rotation which is constrained by the fixture and the plate. The rotational motion of the workpiece can be driven by the LSC fixture or plate. In order to determine its motion, an analysis of the forces on the workpiece in the machining system is required.

Before performing force analysis, we must define that the workpiece, lower plate, and base plate are all ideal planes, so that there is no lubrication between the limiter and the workpiece.

Figure 1 shows the schematic diagram of the force and direction of motion of the workpiece in the planetary double-plane machining system, and Figure 2 shows the force analysis of the workpiece–fixture system during machining. The frictional moment of the workpiece determines the motion of the workpiece rotation.

The friction moment ***M*_O_** exerted by the polishing plate on the surface of the workpiece is:(1)$$\left|M_{O}\right|=\left\{ \begin{array}{l} \frac{2\mu_{O}Fr_{B}}{3}+\mu_{O}mgr_{B}\;\;\;\;\left(\mathrm{lower}\;\mathrm{plate}\right)\\ \frac{2\mu_{O}Fr_{B}}{3}\;\;\;\;\;\;\;\;\;\;\;\;\;\;\;\;\;\;\;\;\;\;\;\left(\mathrm{upper}\;\mathrm{plate}\right) \end{array}\right.,$$
where *μ*_O_ is the coefficient of friction between the plate and workpiece, *m* is the weight of the base plate, ***F***_H_ is the force applied to the base plate by the holder, and *r*_B_ is the radius of the workpiece.

The friction moment ***M*_H_** applied by the holder to the workpiece–fixture system is
(2)$$M_{H}=\mu_{H}F_{H}r_{W},$$
where *μ*_H_ is the coefficient of friction between the holder and the fixture, and *r*_W_ is the radius of the fixture.

The frictional moments generated by the frictional forces between the upper and lower surfaces of the base plate and the workpiece is ***M*_B_**, and the expression is shown in Equation (3).
(3)$$M_{B}=\left\{ \begin{array}{l} \frac{2\mu_{B}r_{B}F}{3}+\mu_{B}\left(m-M\right)gr_{B}\;\;\;\;\;\;\;\;\left(\mathrm{lower}\;\mathrm{plate}\right)\\ \frac{2\mu_{B}r_{B}F}{3}+\mu_{B}Mgr_{B}\;\;\;\;\;\;\;\;\;\;\;\;\;\;\;\;\;\;\;\;\;\left(\mathrm{upper}\;\mathrm{plate}\right) \end{array}\right.,$$
where *μ*_B_ is the coefficient of friction between the baseplate and workpiece, and *M* is the weight of the workpiece.

The frictional moment generated on the workpiece by the action of the upper and lower surface limiters of the fixture is ***M*_xf_**, as shown in Equation (4).
(4)$$M_{xf}=\left\{ \begin{array}{l} \mu_{x}\left[2F\left(\mu_{O}-\mu_{B}\right)+\mu_{O}mg-\mu_{B}\left(m-M\right)g\right]r_{B}\;\;\;\;\;\;\left(\mathrm{lower}\;\mathrm{plate}\right)\\ \mu_{x}\left[2F\left(\mu_{O}-\mu_{B}\right)-\mu_{B}Mg\right]r_{B}\;\;\;\;\;\;\;\;\;\;\;\;\;\;\;\;\;\;\;\;\;\;\;\;\;\;\;\;\;\;\;\;\;\;\left(\mathrm{upper}\;\mathrm{plate}\right) \end{array}\right.,$$
where *μ*_x_ is the coefficient of friction between the limiter and workpiece.

The friction moment of the fixture by the workpiece ***M*_C_** is the sum of the friction moment between the workpiece and the limiter plate and the base plate.
(5)$$M_{C}=M_{B}+M_{xf},$$

The total frictional moment to the workpiece from the polishing plate, the limiter, and the base plate is
(6)$$M_{W}=M_{O}+M_{C},$$

The force of holder on the fixture continuously overcomes the frictional forces exerted by the upper and lower plate, but the frictional moments of the upper and lower plate act on the workpiece by driving the upper and lower surfaces to rotate in their respective directions of motion. We defined *v*_WH’_ as the partial velocity of *v*_WH_ in the directions of *v*_WO_, and *v*_WOU_ and *v*_WOD_ as the linear velocities of the upper and lower plate at the *O*_W_ point, respectively. When min(*v*_WOD_, *v*_WOU_) < *v*_WH’_ < max(*v*_WOD_, *v*_WOU_), the speed of the fixture driven by the holder is between the upper and lower plate speed, and the fixture is subjected to the friction moment exerted by the workpiece on the upper and lower surfaces in the opposite direction, so the fixture is subjected to the combined moment from the workpiece as shown in Equation (7). When *v*_WH’_ > max(*v*_WOD_, *v*_WOU_) or *v*_WH’_ < min(*v*_WOD_, *v*_WOU_), the frictional moments exerted by the upper and lower surfaces on the workpiece shift in the same direction as each other due to the rotational speed, so the fixture is subjected to the combined moment from the workpiece as shown in Equation (7).
(7)$$\left|M_{C}\right|=\left\{ \begin{array}{l} \mu_{B}mgr_{B}\;\;\;\;\;\;\min\left(v_{\mathrm{WOD}},v_{\mathrm{WOU}}\right)<v_{{\mathrm{WH}}^{\prime }}<\max\left(v_{\mathrm{WOD}},v_{\mathrm{WOU}}\right)\\ \left(\left(1-\mu_{x}\right)\mu_{B}-\mu_{O}\mu_{x}\right)\left(2F+mg\right)r_{B}\;\;\;\;\;\;\;\;\left\{ \begin{array}{l} v_{{\mathrm{WH}}^{\prime }}>\max\left(v_{\mathrm{WOD}},v_{\mathrm{WOU}}\right)\\ v_{{\mathrm{WH}}^{\prime }}<\min\left(v_{\mathrm{WOD}},v_{\mathrm{WOU}}\right) \end{array}\right. \end{array}\right.,$$

The frictional moments between the workpiece and the base plate, between the workpiece and the limit plate, and between the holder and the fixture constitute the total frictional moment ***M*_B__H_** to which the fixture is subjected.
(8)$$\left|M_{BH}\right|=\{\begin{array}{c} \mu_{H}r_{B}\sqrt{{\left(2F\mu_{O}+\mu_{O}mg\right)}^{2}+{\left(m\omega_{\mathrm{WB}}^{2}\right)}^{2}}-\mu_{B}mgr_{B}\;\;\;\;\;\;\min\left(v_{\mathrm{WOD}},v_{\mathrm{WOU}}\right)<v_{{\mathrm{WH}}^{\prime }}<\max\left(v_{\mathrm{WOD}},v_{\mathrm{WOU}}\right)\\ \left(\begin{array}{l} \mu_{H}r_{B}\sqrt{{\left(2F\mu_{O}+\mu_{O}mg\right)}^{2}+{\left(m\omega_{\mathrm{WB}}^{2}\right)}^{2}}\\ -\left(\left(1-\mu_{x}\right)\mu_{B}+\mu_{O}\mu_{x}\right)\left(2F+mg\right)r_{B} \end{array}\right)\;\;\;\;\;\;\;\;\;\;\;\;\;\;\;\;\;\;\left\{ \begin{array}{l} v_{{\mathrm{WH}}^{\prime }}>\max\left(v_{\mathrm{WOD}},v_{\mathrm{WOU}}\right)\\ v_{{\mathrm{WH}}^{\prime }}<\min\left(v_{\mathrm{WOD}},v_{\mathrm{WOU}}\right) \end{array}\right. \end{array},$$
where *ω*_WB_ is the angular velocity of rotation of the base plate around the center of the polishing plate.

When |***M*_B__H_**| > 0, the fixture cannot be driven by the workpiece, and the fixture is driven by the holder to produce a revolution with eccentricity *e*_W_ as the radius and a rotation with *O*_W_ as the center of the circle. When |***M*_B__H_**| < 0, the fixture is driven by the frictional moment applied by the workpiece to generate the rotation and revolution. At this point, the critical friction coefficient of the holder to the fixture is obtained according to Equations (9) and (10).
(9)$$\mu_{H1}=\frac{\mu_{B}mg}{\sqrt{{\left(2\mu_{O}F+\mu_{O}mg\right)}^{2}+{\left(m\omega_{\mathrm{WB}}^{2}\right)}^{2}}},\min\left(v_{\mathrm{WOD}},v_{\mathrm{WOU}}\right)<v_{{\mathrm{WH}}^{\prime }}<\max\left(v_{\mathrm{WOD}},v_{\mathrm{WOU}}\right),$$
(10)$$\mu_{H2}=\frac{\left(\left(1-\mu_{x}\right)\mu_{B}+\mu_{O}\mu_{x}\right)\left(2F+mg\right)}{\sqrt{{\left(2F\mu_{O}+\mu_{O}mg\right)}^{2}+{\left(m\omega_{\mathrm{WB}}^{2}\right)}^{2}}},\left\{ \begin{array}{l} v_{{\mathrm{WH}}^{\prime }}>\max\left(v_{\mathrm{WOD}},v_{\mathrm{WOU}}\right)\\ v_{{\mathrm{WH}}^{\prime }}<\min\left(v_{\mathrm{WOD}},v_{\mathrm{WOU}}\right) \end{array}\right.,$$

The total frictional moment of the workpiece is modelled as shown in Equation (11).
(11)$$\left|M_{W}\right|=\left\{ \begin{array}{l} \frac{2Fr_{B}\left(\mu_{O}-\mu_{B}\right)\left(1-3\mu_{x}\right)}{9}+\left(\mu_{B}-\mu_{x}\mu_{B}\right)Mgr_{B}+\left(1-\mu_{x}\right)\left(\mu_{O}-\mu_{B}\right)mgr_{B}\;\;\;\;\;\;\;\left(\mathrm{lower}\;\mathrm{plate}\right)\\ \frac{2Fr_{B}\left(\mu_{O}-\mu_{B}\right)\left(1-3\mu_{x}\right)}{9}-\left(\mu_{B}-\mu_{x}\mu_{B}\right)Mgr_{B}\;\;\;\;\;\;\;\;\;\;\;\;\;\;\;\;\;\;\;\;\;\;\;\;\;\;\;\;\;\;\;\;\;\;\;\;\;\;\;\;\;\;\;\;\;\;\;\;\;\;\;\;\;\;\;\left(\mathrm{upper}\;\mathrm{plate}\right) \end{array}\right.,$$

We define the friction coefficient ratio *k*_OB_ as the ratio of the friction coefficient *μ*_B_ to the friction coefficient *μ*_O_, whose expression is shown in Equation (12).
(12)$$k_{OB}=\frac{\mu_{B}}{\mu_{O}},$$
(13)$$\mu_{x1}=\frac{2F\left(1-k_{\mathrm{OB}}\right)-3k_{\mathrm{OB}}Mg}{6F\left(1-k_{\mathrm{OB}}\right)-3k_{\mathrm{OB}}Mg},$$
(14)$$\mu_{x2}=\frac{2F\left(1-k_{\mathrm{OB}}\right)+3\left(1-k_{\mathrm{OB}}\right)mg+3k_{\mathrm{OB}}Mg}{6F\left(1-k_{\mathrm{OB}}\right)+3\left(1-k_{\mathrm{OB}}\right)mg+3k_{\mathrm{OB}}Mg},$$

Equations (13) and (14) are the critical friction coefficients between the upper surface workpiece and the lower surface workpiece and the limiter, respectively, and it can be seen from the two equations that *μ_x_*_1_ is constantly smaller than *μ_x_*_2_. When *μ_x_ < μ_x_*_1_, the workpiece on the upper and lower surfaces are derived by the plate. When *μ_x_*_1_ < *μ_x_ < μ_x_*_2_, the friction moment of the upper surface workpiece from the plate is not enough to drive the workpiece to overcome the friction moment applied by the fixture, so the upper surface workpiece follows the fixture to rotate, and the lower surface workpiece is driven by the grinding plate to rotate. When *μ_x_ > μ_x_*_2_, the motion of the workpiece is only affected by the orbital revolution and self-rotation of the fixture.

## 3. Analysis and Discussion of Workpiece Motion State

According to previous analysis, the force state of the limiter is mainly determined by the friction coefficient ratio *k*_OB_. When *k*_OB_ > 1, the friction force of the polishing plate on the workpiece is less than the friction force of the base plate on the workpiece. The limiter is not subjected to force and the polishing plate cannot affect the rotational state of the workpiece. Therefore, only cases where *k*_OB_ < 1 are discussed in this analysis. In this analysis, radius of workpiece *r*_B_ = 25.4 mm and mass of workpiece *M* = 0.005 kg were selected. Data values were selected based on actual experimental parameters or enlarged versions of actual experimental data.

The size of the friction force between the limiter and workpiece directly determines the motion state of the workpiece. Therefore, we first analyzed the influencing factors of friction coefficient *μ*_x_ and critical friction coefficient.

We defined *F* = 200 N and *m* = 1.5 kg. Figure 3a shows the relationship between friction coefficient ratio *k*_OB_ and critical friction coefficient. It can be seen from the figure that with a change in friction coefficient, the range of critical friction coefficient values shows a nonlinear change. When *k*_OB_ changes within the range (0, 0.95), the critical friction coefficient between the limiter and workpiece is less affected by *k*_OB_ and is greater than 0.32; it changes sharply within range *k*_OB_ = (0.95, 1). When *k*_OB_ = 1, the value of the critical friction coefficient reaches maximum value and at this time, *μ*_x1_ = *μ*_x2_ = 1. Here, it can be considered that when the friction coefficient *μ*_x_ ≤ 0.32 under current parameters, the fixture cannot drive the workpiece to rotate and the workpiece undergoes self-rotation under the action of friction torque from the polishing plate.

Figure 3b shows the variation curve of the critical friction coefficient taken with the pressure to which the workpiece is subjected at values of *k*_OB_ = 0.9 and *m* = 1.5 kg. From the figure, it can be seen that the lower surface workpiece shows a non-linear decreasing curve due to the gravitational force of the base plate, which eventually converges to 1/3, while the upper surface workpiece shows a more obvious increase at *F* = (0, 10) N. At *F* = 10 N, the critical friction coefficient shows a slow convergence to 1/3. When the rest of the parameters remain unchanged, the greater the applied load, the greater the force on the workpiece, and the frictional torque acting on the workpiece by the polishing plate increases. When the limiter drives the workpiece to rotate, the frictional torque between the workpiece and the polishing plate and the base plate needs to be overcome, so the frictional torque of the limiter on the workpiece needs to increase accordingly, and the frictional torque of the limiter on the workpiece is determined by the friction coefficient. Since the workpiece itself has a very small self-weight, when the load gradually increases, the impact of the self-weight of the workpiece on the critical friction coefficient is negligible. According to the previous analysis, the workpiece on the upper surface of the base plate is only subject to the action of the load *F*. Therefore, the critical friction coefficient u_x1_ increases with the increase of the load and tends to 1/3. The workpiece on the lower surface of the base plate is much larger than the self-weight of the workpiece due to the action of the self-weight of the base plate m. Therefore, when the load is 0, the critical friction coefficient *μ*_x2_ takes on a larger value, and when the load increases, the effect of *m* on the critical friction coefficient decreases. The value of the critical friction coefficient *μ*_x2_ shows a gradually decreasing trend. The above results show that when the force on the surface of the workpiece is greater than 35 N and the friction coefficient *μ*_x_ ≤ 0.33, the workpiece is subjected to self-transmission by the frictional moment exerted on it by the polishing plate.

According to Equations (9) and (10), the friction coefficient *μ*_H_ is influenced by the friction coefficients *μ*_x_, *μ*_O_, *μ*_B_, and *ω*_WB_, where the direction of *ω*_WB_ has no effect on the critical value of *μ*_H_. Defining *ω*_WB_ = 30 rpm can obtain two critical values of *μ*_H_ at the same time.

Figure 4a shows the relationship between pressure *F*, mass *m*, and critical friction coefficient *μ*_H1_ where *μ*_x_ = 0.3, *μ*_O_ = 0.5, and *μ*_B_ = 0.2. It can be seen from the figure that when the rotational speed of the workpiece is between the speeds of the upper and lower plates, external force F and mass m of base plate have almost no effect on *μ*_H1_, with the maximum value being about 0.0025. Figure 4b shows the curve of change in *μ*_H2_ with *F* and mass *m*. When *F* is constant, the critical friction coefficient increases with an increase in mass; when mass is a constant value, *μ*_H2_ decreases nonlinearly with increase in *F*. When mass m of base plate is 2 kg and applied pressure *F* > 18.3 N, critical friction coefficient is less than 0.1.

Figure 5 shows the relationship between friction coefficients *μ*_O_ and *μ*_B_ and critical friction coefficient *μ*_H_ when *m* = 2 kg, *F* = 100 N, and *μ*_x_ = 0.3. Figure 5a shows the distribution of the critical friction coefficient *μ*_H1_ when the rotational speed *ω*_WB_ of the base plate is between the speeds of the upper and lower plates. From the distribution results, when friction coefficients *μ*_O_ and *μ*_B_ change within range (0, 1), the maximum value of the critical friction coefficient *μ*_H1_ obtained is 0.0109. Figure 5b shows the distribution of the critical friction coefficient when the rotational speed *ω*_WB_ of the workpiece is greater than or less than the speeds of the upper and lower plates. From the results, when *μ*_O_ is constant, an increase in *μ*_B_ promotes an increase in μ_H2_ but the magnitude of increase is small; when *μ*_B_ is constant, *μ*_H2_ increases with an increase in *μ*_O_, and compared to *μ*_B_, the influence of *μ*_O_ on *μ*_H2_ is greater. It should be particularly noted that when both *μ*_O_ and *μ*_B_ are less than 0.8, the maximum value of *μ*_H2_ is 0.0976. Analyzing the contact state between the holder and base plate shows more sliding friction than rolling friction. Therefore, it is judged that when critical friction coefficients *μ*_H1_ and *μ*_H2_ < 0.1, the sliding friction force between the holder and base plate is always greater than the critical friction coefficients *μ*_H1_ and *μ*_H2_. At this time |***M***_BH_| > 0, the base plate is driven by the holder to generate revolution movement with eccentric distance *e*_w_ as radius and self-rotation movement with *O*_W_ as the center.

We defined *m* = 2 kg, *F* = 100 N, *μ*_x_ = 0.3, *μ*_O_ = 0.5, *μ*_B_ = 0.2, and lower polishing plate speed *ω*_OD_ = 40 rpm. Figure 6 shows the influence of rotational angular velocity *ω*_WB_ on critical friction coefficient value at different upper polishing plate speeds (*ω*_OU_ = 5 in (a), *ω*_OU_ = −15 in (b), and *ω*_OU_ = −35 in (c)). It can be seen from Figure 6a that when changing between (−200, 0), the value of the critical friction coefficient increases with an increase in rotational angular velocity *ω*_WB_ and reaches a maximum value of 0.58 when *ω*_WB_ = 0. Then, it decreases with an increase in ω_WB_. When *ω*_WB_ is between the speeds of the upper and lower plates, its critical value is small and the maximum value is obtained when *ω*_WB_ = 5 rpm with a value of 0.032. When *ω*_WB_ is greater than the speed of the lower plate, the critical friction coefficient decreases with an increase in *ω*_WB_. Figure 6b,c show that when the upper disc speed *ω*_OU_ decreases, the maximum value of the critical friction coefficient changes. This is particularly evident in Figure 6c, where the maximum value of the critical friction coefficient *μ*_H_ is 0.0357.

Due to characteristics of planetary structure, the rotational angular velocity of the base plate is not constant and its rotational angular velocity changes cyclically. Analyzing Figure 6 shows that when the workpiece is processed at the lower upper disc speed, the change in rotational angular velocity of the base plate easily causes the critical friction coefficient to have a larger value. At this point, the base plate is in a mixed driving state driven by the workpiece and holder. The motion speed of the base plate easily produces sudden stop and sudden turn phenomena and should be avoided as much as possible. Therefore, the speed of the upper and lower polishing plates should avoid taking smaller speeds as much as possible. Combining Figure 6b,c shows that when polishing plate speed is greater than 15 rpm, a smaller critical friction coefficient can be obtained to ensure that there is only a single power source during the motion of the base plate.

## 4. Experimental Research and Discussion

In order to analyze frictional behavior between the workpiece and base plate under LSC with different material base plates and roughnesses, an experimental research method was used for analysis. This experiment uses the multi-dimensional force measurement platform shown in Figure 7. The force sensor was a three-dimensional force sensor produced by ME-Meßsysteme with a measurement accuracy of 0.02 N. The experimental object was a sapphire wafer with a diameter of ϕ50.8 mm. The flatness of the sapphire wafer was 3.4 μm, surface roughness was *R*_a_ = 5 nm, and thickness was 170 μm. The height difference on the surface of the base plate used was 10 μm. The contact state between the workpiece and base plate was the wet state. One drop of deionized water was titrated on the surface using a 5 mL burette. Relevant experimental parameters are shown in Table 1. When conducting experiments on base plates of different materials, the same level of roughness was selected for measurement.

### 4.1. Tangential Force Analysis of Base Plate

Comparing with Figure 8, it can be seen that under the water film adsorption state, cast iron and stainless-steel materials have different degrees of friction force increase compared to the dry friction state. Contact deformation between sapphire wafer and rough surface forms larger plowing and adhesion effects, thereby generating sliding friction force. In the state of a small amount of liquid adsorption, the water medium is introduced between different solid materials. An extremely thin water film is formed on micro-convex peaks of rough surface. During the lateral sliding process, the adhesion force inside water and the adhesion force between the water, workpiece, and base plate increase resistance during the tangential movement process, thereby generating greater friction force between two surfaces.

Figure 9 shows the friction force under adsorption and dry friction conditions for the same material with different surface roughness. It can be seen from the figure that under dry friction conditions, the friction coefficient remains relatively stable as roughness increases. Among them, the friction coefficient is smaller when *R*_a_ = 3.6 nm because the surface is smoother and cannot form an effective adhesion and plowing effect, thereby reducing tangential force between two surfaces and reducing the friction coefficient. Under adsorption conditions, however, the friction coefficient first increases then decreases as roughness increases because when the workpiece is relatively smooth, the additional adhesion force caused by water adsorption and the lubrication effect of water on the surface have corresponding offset effects. At this time, the change in friction coefficient caused is relatively small. For surfaces with larger roughness, the additional adhesion force brought by water adsorption on the surface is relatively small and the proportion of the lubrication effect of water increases. The friction coefficient decreases instead.

Combining analysis of Figure 8 and Figure 9 shows that when the base plate material is stainless steel and the surface roughness of the base plate is 68.2 nm, the maximum friction coefficient can be obtained. At this time, the average value of friction coefficient is 0.173.

### 4.2. Materials of Base Plate and Limiter

The deformation of the limiter under radial force with elastic deformation is Δ*r*, the clamping thickness of the limiter is Δ*h*, and the contact area with the workpiece at this time is:(15)$$S=2r_{B}\Delta h\arccos\frac{r_{B}-\Delta r}{r_{B}},$$

The shear strength *τ* of the limiter is:(16)$$\tau =\frac{\left|F_{xD}\right|}{S}=\frac{\left(F+mg\right)\left(\mu_{O}-\mu_{B}\right)+\mu_{B}Mg}{2r_{B}\Delta h\arccos\frac{r_{B}-\Delta r}{r_{B}}},$$

We defined the width of the limiter as *q*, and therefore the strain between the workpiece and the limit piece at this time was *ε =* Δ*r/q*. According to the material strain and stress formula, the solution formula of Δ*r* can be obtained as follows:(17)$$\Delta r=\frac{\tau q}{E},$$
where *E* is the material modulus of elasticity.

Considering installation issues, the workpiece and limiter are clearance fit, so there is fatigue load in actual processing. We define calculated stress as *τ*_ca_ = *τ/S*, where *S* is safety factor, and radius of workpiece as *r*_B_ = 25.4 mm, weight of base plate and workpiece *m* = 2.1 kg, mass of workpiece *M* = 0.003 kg, and clamping thickness Δ*h* = 0.1 mm. When fiberglass board is used as the limiter material, the elastic modulus of fiberglass is *E* = 38 GPa and ultimate shear strength is 53.9 MPa. Safety factor *S* is defined as 1.3. Combined with Equation (17), the limit elastic deformation of limiter Δ*r* = 21.2763 μm and load that the limiter can withstand is *F*_xD_ = 86.216 N.

According to the analysis in Figure 8, the friction coefficient *μ*_B_ between the workpiece and base plate during the clamping stage are: aluminum alloy *μ*_B_ = 0.073, stainless steel *μ*_B_ = 0.1830, cast iron *μ*_B_ = 0.1493. The base plate should be selected from materials with larger friction force to reduce requirements for the limiter, so stainless steel material was selected for base plate preparation. Since there is polishing liquid between the workpiece and polishing pad during processing, polishing liquid has a good lubrication effect between the workpiece and polishing pad at relatively high linear speed. Considering reliability, *μ*_O_ = 0.5 is defined in this analysis. At this time, the maximum pressure on the workpiece is *F*_max_ = 287.4 N.

### 4.3. Experimental of Limiter Clamping Thickness

The clamping thickness of the limiter has an important impact on the processing thickness of ultra-thin sapphire. As can be seen from the research in the previous section, when stainless steel and fiberglass board are selected as the base plate and limiter materials for laminated ultra-thin sapphire, the maximum pressure that can be withstood is 287.4 N. In order to compare the reliable thickness of limiter clamping, pressure is applied to limiters of different thicknesses to test the maximum processing load that can be withstood at that thickness. Therefore, in this experiment, the clamping thickness Δ*h* of the limiter was tested at four levels, with values of 0.081, 0.102, 0.121, and 0.151 mm, respectively; SiO_2_ polishing liquid with particle size of 80 nm was used with a concentration of 5% and pH value of 12.2. Specific test parameters are shown in Table 2. To ensure the reliability of the experiment, a Nanopoli-100 single-sided polishing machine, shown in Figure 10, was used in this experiment. The diameter of the plate was ϕ300 mm, polishing speed was 60 rpm, and each pressurized processing time was 1 h.

The experimental results are shown in Table 3. It can be seen that as the clamping thickness increases, the limit value of the normal force that the limiter can withstand also increases.

Figure 11 shows the failure form of the limiter. It can be seen from the figure that the failure state of the limiter is that the limit area is cut by the edge of the sapphire to form a slope. During processing of sapphire, due to changes in flatness of the base plate and sapphire wafer, the wafer and base plate are not parallel at some stage of processing, causing the wafer to warp relative to the base plate. The friction force applied by the polishing pad on the workpiece drives the workpiece to squeeze the limit area. The contact area between the limit area and sapphire wafer decreases, causing shear strength in the limit area to increase sharply. At the same time, the sapphire wafer has self-rotating angular velocity. When the raised edge rotates, it will cut the limiter. Under the combined action of increased shear force and cutting action, the structure of the limiter is damaged. The sapphire wafer slips out of the LSC fixture and eventually causes fragments.

## 5. Conclusions

In order to explore the motion state of ultra-thin sapphire wafers under the LSC clamping method, a motion state analysis model was established and the influence of various factors on the motion state was analyzed. Through experimental research on the materials of the limiter and base plate, and determination of the failure form of the limiter, the following conclusions were obtained:

The motion state of the workpiece in the LSC fixture is mainly affected by the friction coefficient ratio *k*_OB_ and the friction coefficient between the workpiece and the limiter. When *k*_OB_ < 1 and friction coefficient *μ*_x_ ≤ 0.33, the workpiece is driven by the polishing plate to rotate, and the motion state of the workpiece is independent of the motion state of the base plate and limiter.When the speed of the upper and lower polishing plates is greater than 15 rpm and directions are opposite, the motion of the base plate is always driven by the holder.Sliding friction experiments show that as surface roughness of the base plate increases, due to the adhesion of water film, the friction force between the workpiece and base plate first increases then decreases. Stainless steel is the preferred material for the base plate.Based on material mechanics, the material for limiter was selected as fiberglass board. The failure form of the limiter was determined through experiments. It manifested as being subjected to rotating cutting action of sapphire wafer, causing the limiter to form an inclined surface and the workpiece to slip and break.

## Figures and Tables

**Figure 1 micromachines-14-01124-f001:**
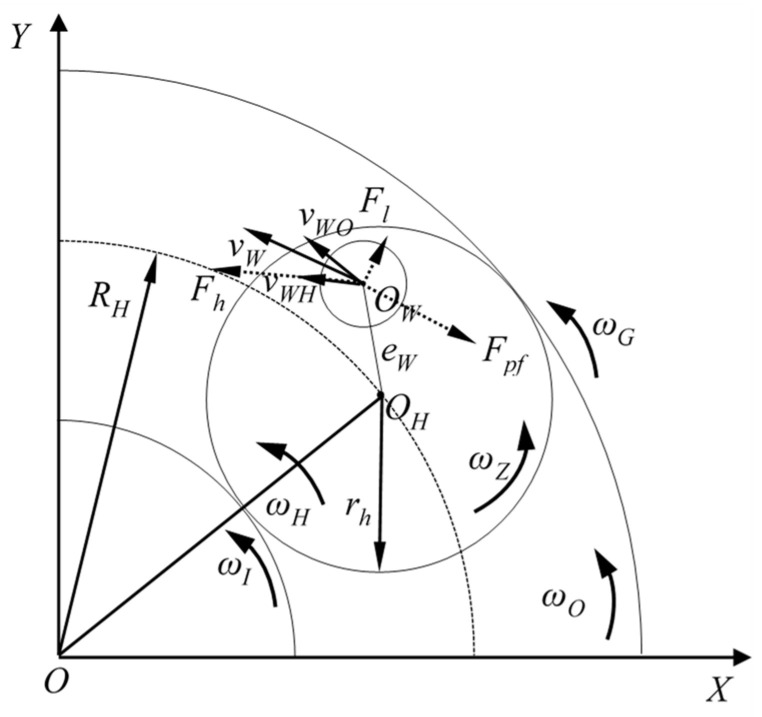
Analysis of the motion state of the workpiece.

**Figure 2 micromachines-14-01124-f002:**
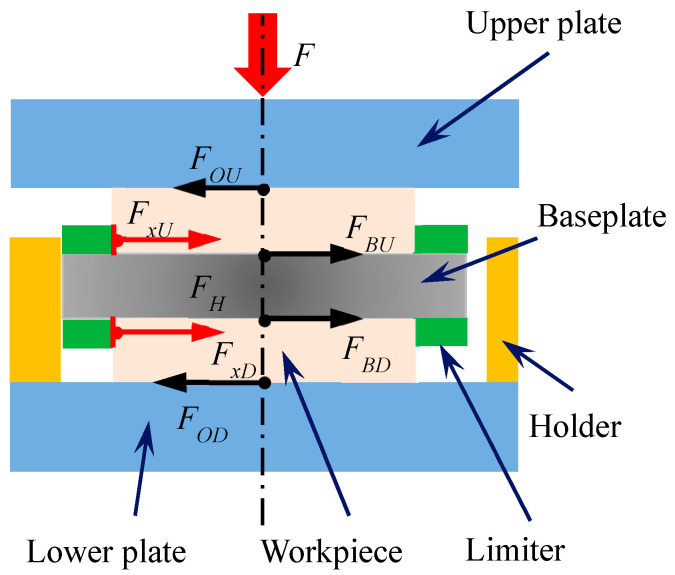
Force analysis of workpiece–fixture system.

**Figure 3 micromachines-14-01124-f003:**
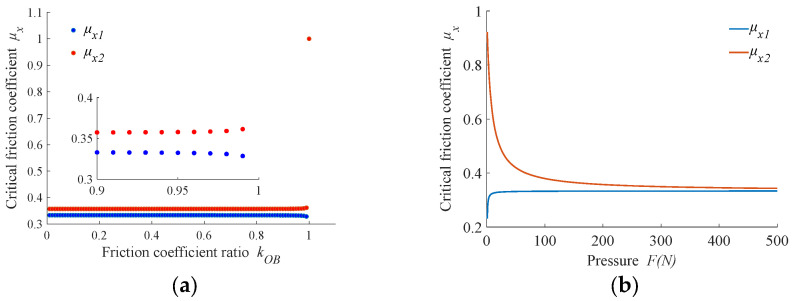
Variation of the critical friction coefficient *μ*_x_: (**a**) influence from friction coefficient ratio *k*_OB_; (**b**) influence from pressure *F*.

**Figure 4 micromachines-14-01124-f004:**
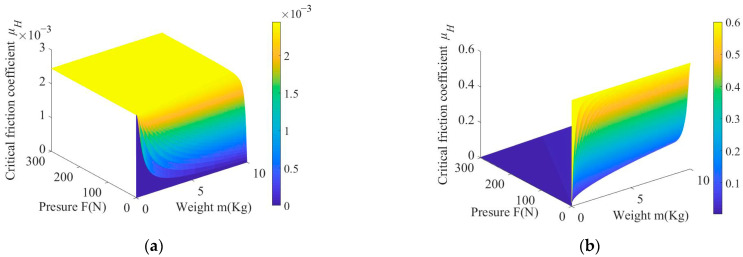
Relationship between pressure *F* and mass *m* and critical friction coefficient *μ*_H_: (**a**) critical friction coefficient *μ*_H1_; (**b**) critical friction coefficient *μ*_H2_.

**Figure 5 micromachines-14-01124-f005:**
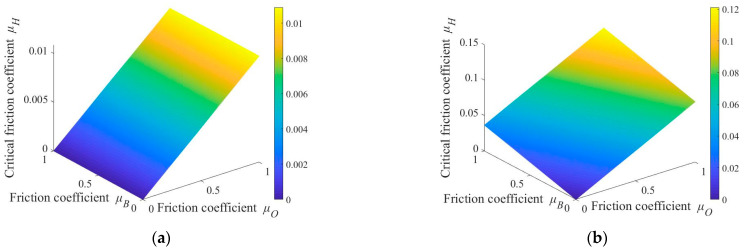
Relationship between friction coefficient *μ*_O,_*μ*_B_ and critical friction coefficient *μ*_H_: (**a**) critical friction coefficient *μ*_H1_; (**b**) critical friction coefficient *μ*_H2_.

**Figure 6 micromachines-14-01124-f006:**
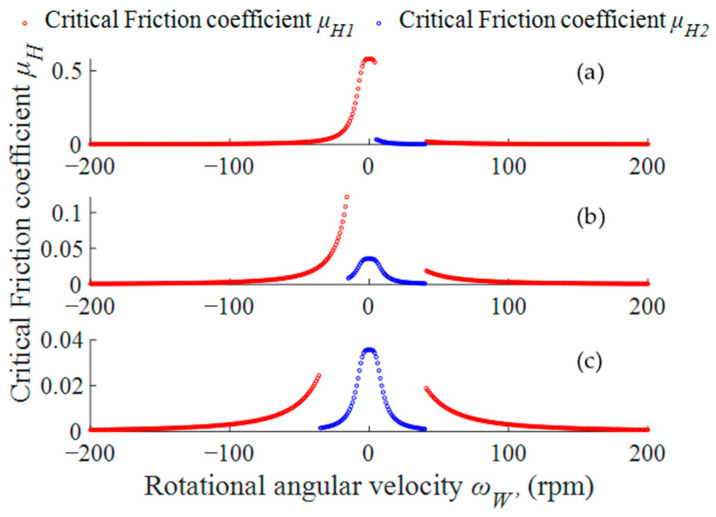
Effect of rotational angular velocity *ω*_WB_ on critical friction coefficient. (**a**) *ω*_OU_ = 5 rpm; (**b**) *ω*_OU_ = −15 rpm; (**c**) *ω*_OU_ = −35 rpm.

**Figure 7 micromachines-14-01124-f007:**
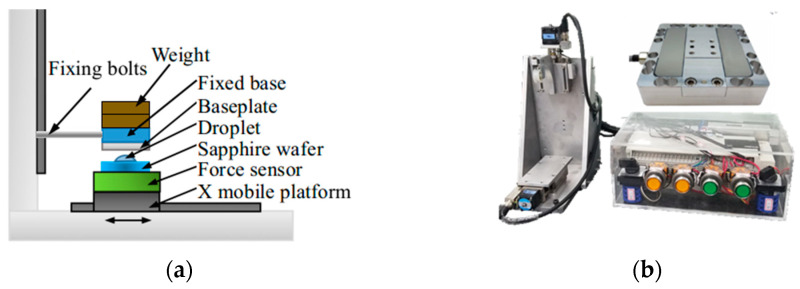
Friction experiment platform: (**a**) measurement principle; (**b**) equipment and sensors.

**Figure 8 micromachines-14-01124-f008:**
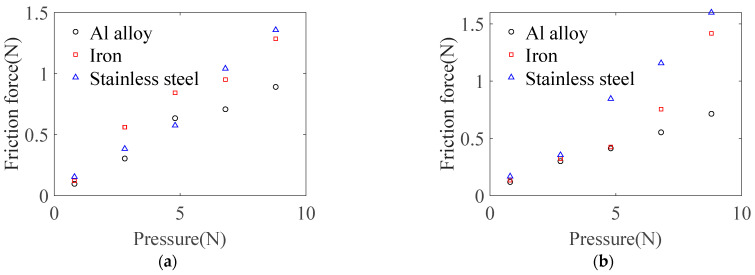
Sliding friction between different material base plate and sapphire wafers: (**a**) dry friction; (**b**) wetting friction.

**Figure 9 micromachines-14-01124-f009:**
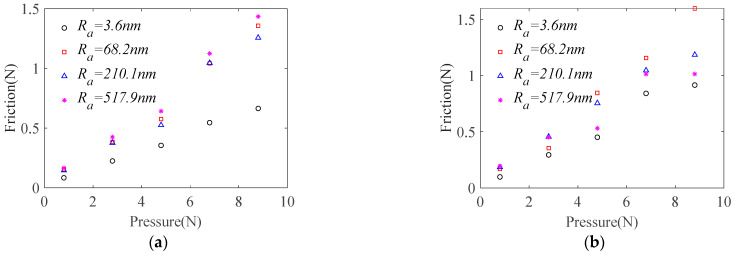
Sliding friction between different roughnesses of base plate and sapphire wafers: (**a**) dry friction; (**b**) wetting friction.

**Figure 10 micromachines-14-01124-f010:**
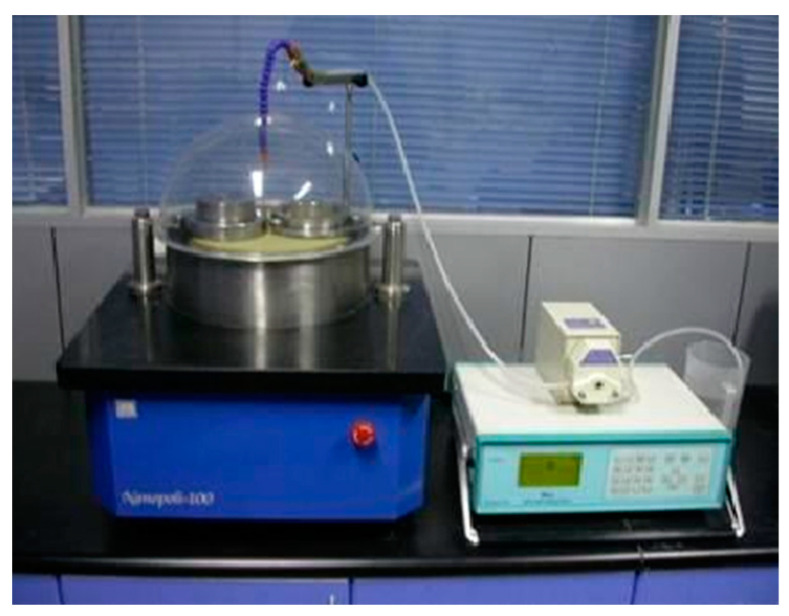
Single-sided polishing machine (Nanopoli-100).

**Figure 11 micromachines-14-01124-f011:**
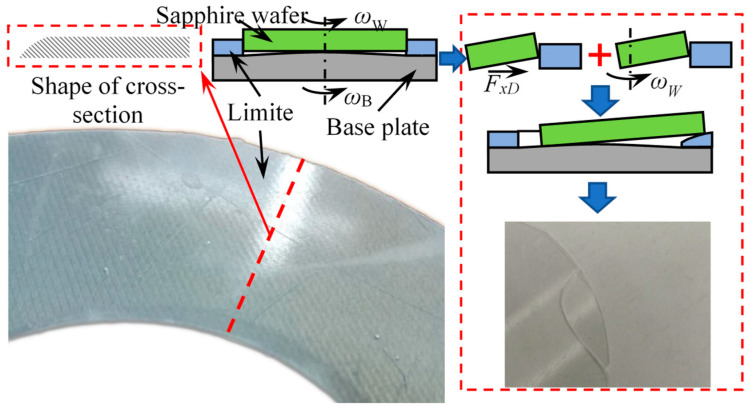
The failure mode and cause of the limiter plate.

**Table 1 micromachines-14-01124-t001:** Friction experiment parameters.

Parameters		Value
Pressure (N)		0.8, 2.8, 4.8, 6.8, 8.8
Movement speed of *X*-axis (mm/s)		0.1
Material of base plate		Stainless steel, Aluminum alloy, Cast iron
Roughness of base plate (nm)	Stainless steel	3.6, 68.2, 210.1, 517.9
	Aluminum alloy	60.2
	Cast iron	63.8

**Table 2 micromachines-14-01124-t002:** Experimental parameters.

Parameters	Value
Height difference of base plate (μm)	10
Quality of base plate (kg)	2.1
Weights (N)	6.86
Number of weights	1–15
Flow rate of polishing slurry (L/h)	1500

**Table 3 micromachines-14-01124-t003:** Experimental results.

Limiter Thickness (mm)	Damaged Pressure (N)
0.082	48.02
0.104	68.6
0.119	96.04
0.151	Undamaged

## Data Availability

The data presented in this study are available on request from the corresponding author. The data are not publicly available due to privacy.

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
