# Peer review of "Analysis of the Influence of the Motion State of Ultra-Thin Sapphire Based on Layer-Stacked Clamping (LSC)"

_micromachines, 2023, doi:10.3390/mi14061124_

Round 1

Reviewer 1 Report

In this paper, the influence of the motion state under double-sides polishing of sapphire based on layer stacked clamping was investigated by experiments and theoretical analysis. The results gives the basis for judging the motion state of the workpiece and the base plate, and lays the foundation for the subsequent workpiece surface trajectory uniformity evaluation. A lot of work done by the authors is worth encouraging, and this paper meets the requirement of Micromachines. There are some defects in the paper that need to be considered and corrected.

(1) Section 3, line 149, figure 3(b), please explain why μx1 keeps decreasing and μx1 keeps increasing.

(2) Section 3, line 180, figure 4(a), the gridlines in Figure 4a should be consistent with those in Figure 4b, please adjust the other pictures in the manuscript accordingly.

(3) Section 3, line 212, figure 6, please explain why the line segments are not continuous.

(4) Note the notation of the variables in the text and their descriptions, such as Ra in line 254 and μB in line 287.

(5) Section 4.2, line 286, the author mentioned Figure 5 3, which was not found in the text, please explain this.

(6) The following papers also studied the precision machining of hard and brittle wafers, which are related to your work. The authors can refer to them in the manuscript.

[A] Phase transition and plastic deformation mechanisms induced by self-rotating grinding of GaN single crystals. International Journal of Machine Tools and Manufacture, 2022, 172: 103827.

[B] Molecular dynamics simulation of laser assisted grinding of GaN crystals. International Journal of Mechanical Sciences. 2023, 239, 107856.

Author Response

Point 1: Section 3, line 149, figure 3(b), please explain why μx2 keeps decreasing and μx1 keeps increasing.

Response 1: Thank you. Some explanation have been added to section 3,line 164-180,“When the rest of the parameters remain unchanged……The value of the critical friction coefficient μx2 shows a gradually decreasing trend.”

Point 2: Section 3, line 180, figure 4(a), the gridlines in Figure 4a should be consistent with those in Figure 4b, please adjust the other pictures in the manuscript accordingly.

Response 2: Thank you. The author has adjusted the corresponding picture.

Point 3: Section 3, line 212, figure 6, please explain why the line segments are not continuous.

Response 3: Thank you. According to Eq. 9-10, when the speed of vWH' changes, the critical friction coefficient uh will change with the change of vWH', when vWH'=max(|vWOD|, |vWOU|) or min(|vWOD|, |vWOU|), at this time the critical friction coefficient μH is in the critical change state, that is, in μH2 and μH1 critical state change. Therefore, a discontinuity occurs at different rotational speeds. Section 3, line 231, Figure 6 has been redrawn for ease of understanding.

Point 4: Note the notation of the variables in the text and their descriptions, such as Ra in line 254 and μB in line 287.

Response 4: Thank you. The author has revised the variable marking and description in the manuscript accordingly.

Point 5: Section 4.2, line 286, the author mentioned Figure 5 3, which was not found in the text, please explain this.

Response 5: Thank you. Due to the mistake, the figure number was wrong. The corresponding figure has been corrected, and the corrected figure is shown in Figure 8.

Point 6: The following papers also studied the precision machining of hard and brittle wafers, which are related to your work. The authors can refer to them in the manuscript.

[A] Phase transition and plastic deformation mechanisms induced by self-rotating grinding of GaN single crystals. International Journal of Machine Tools and Manufacture, 2022, 172: 103827.

[B] Molecular dynamics simulation of laser assisted grinding of GaN crystals. International Journal of Mechanical Sciences. 2023, 239, 107856.

Response 6: Thank you. The references were cited as ref. 19-20 in introduction according to the comment.

Reviewer 2 Report

In this study, an analytical model of the motion state of sapphire wafers in the layer stacked clamping process based on the frictional moment is established, the effect of each friction coefficient on its motion is discussed, the base plate of different materials and different roughness are experimentally studied, and the layer stacked clamping fixture is prepared in this way, and finally the failure form of the limiting tab is analyzed experimentally. The theoretical analysis shows that the sapphire wafer is mainly driven by the polishing plate, while the base plate is mainly driven by the holder, and the rotation speed of the two is not the same. Therefore, the paper is recommended for publication in the Micromachines after the following questions can be addressed.

1. Delete a “Abstract:” of line 10.

2. The entire manuscript needs to verify for grammatical and typo errors.

3. Please give the structure of this paper at the end of the introduction.

4. All formulas are followed by punctuation.

5. “Figure 9” in this section of line 200 is changed to “Figure 6”.

6. “Figure 5 3” of line 286 and “Table 5.2” of line 304 are wrong, please modify them.

7. The volume and page numbers of literature 11 should be updated.

Minor editing of English language required.

Author Response

Point 1: Delete a “Abstract:” of line 10.

Response 1: Thank you. ‘Abstract:’ was deleted according to the comment.

Point 2: The entire manuscript needs to verify for grammatical and typo errors.

Response 2: Thank you. Authors have tried best to revise grammatical and typo errors.

Point 3: Please give the structure of this paper at the end of the introduction.

Response 3: Thank you. The structure of manuscript has given in line 66-72.

Point 4: All formulas are followed by punctuation.

Response 4: Thank you. the punctuation was added in all formulas according to the comment.

Point 5: “Figure 9” in this section of line 200 is changed to “Figure 6”.

Response 5: Sorry for the mistake. It has been corrected according to the comment.

Point 6: “Figure 5 3” of line 286 and “Table 5.2” of line 304 are wrong, please modify them.

Response 6: Sorry for the mistake. It has been corrected according to the comment. Added the missing image, as shown in Figure 10, line 327

Point 7: The volume and page numbers of literature 11 should be updated.

Response 7: Thank you. Authors have tried best to revise references format.

Round 2

Reviewer 2 Report

It is OK!